# Germinal center B cells that acquire nuclear proteins are specifically suppressed by follicular regulatory T cells

Fang Ke[1], Zachary L Benet[1], Mitra P Maz[2], Jianhua Liu[2], Alexander L Dent[3], Joanne Michelle Kahlenberg[2], Irina L Grigorova[1]*†

[1]Department of Microbiology and Immunology, University of Michigan–Ann Arbor, Ann Arbor, United States; [2]Department of Internal Medicine, Division of Rheumatology, University of Michigan–Ann Arbor, Ann Arbor, United States; [3]Department of Microbiology and Immunology, Indiana University School of Medicine, Indianapolis, United States

**Abstract** Follicular regulatory T cells (Tfr) restrict development of autoantibodies and autoimmunity while supporting high-affinity foreign antigen-specific humoral response. However, whether Tfr can directly repress germinal center (GC) B cells that acquire autoantigens is unclear. Moreover, TCR specificity of Tfr to self-antigens is not known. Our study suggests that nuclear proteins contain antigens specific to Tfr. Targeting of these proteins to antigen-specific B cells in mice triggers rapid accumulation of Tfr with immunosuppressive characteristics. Tfr then exert negative regulation of GC B cells with predominant inhibition of the nuclear protein-acquiring GC B cells, suggesting an important role of direct cognate Tfr-GC B cells interactions for the control of effector B cell response.

### Editor's evaluation

It is well known that Tfr cells have the capacity to preferentially suppress autoimmune antibody responses, but it is not known why such specificity is generated. This important study provides new information as to how self-reactive antibody responses are regulated and has significant implications for the fields of autoimmunity and vaccine design. The authors added careful controls which are convincing enough.

**\*For correspondence:**
igrigor@umich.edu

†Lead contact

## Introduction

Follicular regulatory T cells (Tfr) are a subset of regulatory T cells, predominantly CXCR5[high]PD1[high]-FoxP3[+], that are present in B cell follicles of secondary lymphoid organs (SLOs) (*Chung et al., 2011*; *Lim et al., 2004*; *Linterman et al., 2011*; *Wollenberg et al., 2011*). Tfr play multiple roles in the regulation of B cell responses. On one side, Tfr prevent development of auto-Abs and autoimmunity (*Botta et al., 2017*; *Clement et al., 2019*; *Fu et al., 2018*; *Gonzalez-Figueroa et al., 2021*; *Wu et al., 2016*) and in some studies appear to have a modest negative effect on the GC and Ab response (*Chung et al., 2011*; *Clement et al., 2019*; *Fu et al., 2018*; *Sage et al., 2016*; *Wollenberg et al., 2011*). In addition to that, Tfr affect B cell recruitment and selection in germinal centers (GCs) (*Cavazzoni et al., 2022*; *Chung et al., 2011*; *Clement et al., 2019*; *Laidlaw et al., 2017*; *Lim et al., 2004*; *Linterman et al., 2011*; *Lu et al., 2021*; *Sage et al., 2016*; *Wollenberg et al., 2011*; *Wu et al., 2016*; *Xie et al., 2020*).

GC B cell responses critically depend on the provision of help by Tfh cells. Tfh cells differentiate from foreign Ag-specific Th cells following immunization or infection. Importantly, differentiation of both Tfh cells and Tfr depends on the molecular interactions with antigen-presenting cells, involving SAP and ICOS molecules and the master transcription factor BCL6 (*Hu et al., 2013*; *Johnston et al., 2009*; *Linterman et al., 2011*; *Nurieva et al., 2009*; *Pedros et al., 2016*; *Sage et al., 2013*; *Xu et al., 2013*). Moreover, accumulation of Tfr in SLOs occurs at the same time or slightly after the Tfh cells. However, in contrast to Tfh cells, the Tfr TCR repertoire predominantly overlaps with natural FoxP3+ Tregs and is more diverse than the Tfh TCR repertoire (*Maceiras et al., 2017*). While some studies suggest that TCR signaling must be important for Tfr development (*Maceiras et al., 2017*; *Shrestha et al., 2015*), which self-antigens are cognate to Treg-derived Tfr is largely unknown. Such analysis is further complicated by very limited knowledge of the natural immunodominant self-antigens cognate to thymus derived Tregs (*Leonard et al., 2017*). The deficiency in the identification of Tfr-specific self-antigens makes it difficult to assess whether Tfr may form direct cognate interactions with MHCII/self-peptides on the GC B cells.

Some experimental in vivo evidence supports the role of direct Tfr interactions with GC B cells for the negative regulation of immune responses. Tfr-produced membrane-attached neuritin has been recently shown to be acquired by B cells leading to their reduced ability to form plasma cells (*Gonzalez-Figueroa et al., 2021*). In addition, our previous study suggested that GC B cell-intrinsic production of CCL3 promotes direct contacts with Tfr and modest inhibition of the foreign Ag-specific B cells at the peak of GC response (*Benet et al., 2018*). However, the contribution of direct Tfr-GC B cell encounters to the suppression of self-antigen-acquiring and potentially autoreactive GC B cells is not known. The alternative hypothesis is that Tfr may restrict autoreactive B cell responses nonspecifically by reducing B cell activation/selection threshold through general inhibitory actions on B and/or Tfh cells.

In this study we show that nuclear proteins (NucPrs), often targeted by auto-Abs in autoimmune diseases, contain ligands that could trigger significant accumulation of Tfr. Targeting these NucPrs to antigen-specific B cells by booster immunization induces rapid accumulation of Tfr with elevated expression of immunosuppressive genes. Activated Tfr then promote modest inhibition of the total GC responses with predominant suppression of the NucPr-acquiring GC B cells, memory B cells, and plasma cells. These findings suggest an important role of direct interactions between GC B and Tfrs that specifically recognize self-antigen acquired and presented by GC B cells in the immunoregulation.

## Results

### Acquisition of NucPrs by GC B cells promotes rapid accumulation of Tfr

Tfr deficiency leads to accumulation of anti-nuclear and tissue-specific antibodies (Abs) (*Gonzalez-Figueroa et al., 2021*), leading to development of auto-Ab-mediated autoimmunity in older mice (*Fu et al., 2018*). NucPrs that are often targeted in autoimmune diseases by auto-Abs include nucleosomal histones, SS-A/Ro, RNP-Sm (small nuclear ribonucleoproteins), Scl70, and Jo-1 (aminoacyl-tRNA synthetase, both nuclear and cytoplasmic) (*Mahler and Fritzler, 2010*; *Tan, 1983*). In this study we hypothesized that these NucPrs may contain peptides cognate to Tfr. Because the rise in Tfr is usually observed at the same time or slightly after the increase in Tfh/GC B cells (*Aloulou et al., 2016*; *Turner et al., 2017a*), we first examined whether acquisition of NucPrs by GC B cells promotes accumulation of Tfr (*Figure 1A*). Bovine nucleosomes, SSA-Ro, RNP-Sm, Scl70, and Jo1 (that have over 92% homology to murine NucPrs), were biotinylated and conjugated to streptavidin (SA) to generate SA-NucPrs. Alternatively, SA was conjugated to foreign antigens (Ags): ovalbumin-biotin or duck egg lysozyme (DEL)-biotin to generate SA-OVA or SA-DEL Ags respectively. C57BL/6 (B6) mice were subcutaneously immunized with SA-DEL to recruit both SA and DEL-specific B cells into the GC response. After formation of GCs (8 days later) mice were boosted with SA-NucPrs, SA, or SA-OVA to promote acquisition of these Ags by the SA-specific GC B cells. The Ag-draining inguinal lymph nodes (dLNs) were collected 3 days later for flow cytometry analysis. We found that Tfr in the dLNs of SA-NucPrs-boosted mice were significantly increased as compared to mice boosted with SA or SA-OVA Ags (*Figure 1A–C*). In contrast, the numbers and percentage of Tfh cells in the SA-NucPrs-boosted mice slightly decreased (*Figure 1D and E*). The observed accumulation of CXCR5$^{high}$ PD1$^{high}$ FOXP3$^+$ Tfr was not accompanied by increase in the frequencies of CXCR5$^{int}$ PD1$^{int}$ FOXP3$^+$ or CXCR5$^{low}$ PD1$^{low}$

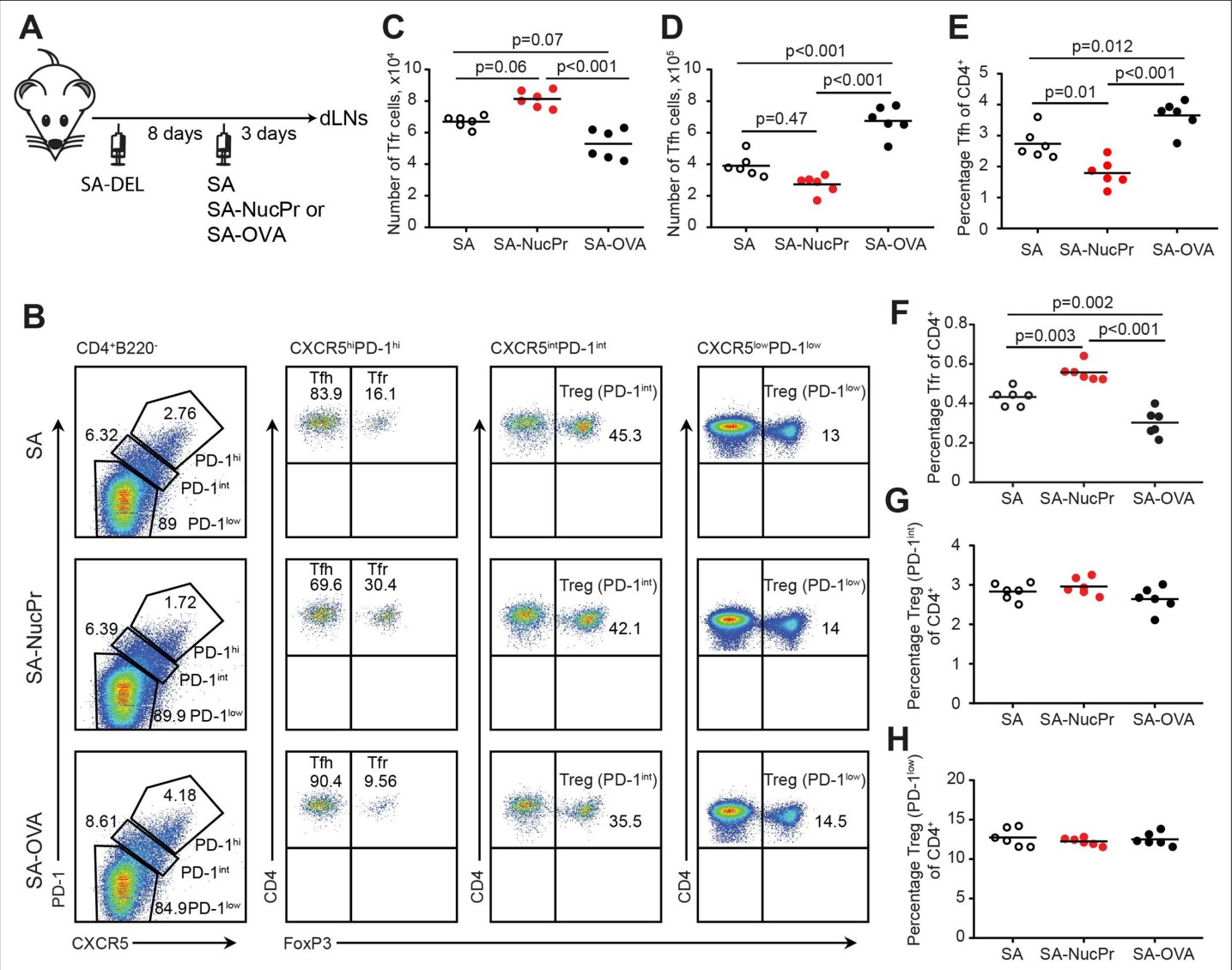

**Figure 1.** Boosting mice with streptavidin (SA) linked to nuclear proteins (NucPrs) induces rapid follicular regulatory T cell (Tfr) response. (**A**) Experimental outline. B6 mice were subcutaneously (s.c.) immunized with SA-DEL in Ribi and at day 8 were s.c. reimmunized with SA, SA-NucPr, or SA-OVA in Ribi for analysis 3 days later. (**B–G**) Flow cytometry analysis of Tfr, Tfh, and other Treg subsets in the draining inguinal lymph nodes (dLNs) of immunized mice. (**B**) The gating strategy to identify the CXCR5hiPD1hi FOXP3+ (Tfr) and CXCR5hiPD1hi FOXP3- (Tfh), CXCR5intPD1int FOXP3+ (PD1int Tregs), and CXCR5lowPD1low FOXP3+ (PD1low Tregs) cell populations and representative flow plots for SA, SA-NucPr, and SA-OVA-boosted mice. (**C**) The numbers of Tfr. (**D, E**) The numbers (**D**) and percentage (**E**) of Tfh cells. (**F–H**) The frequencies of Tfr (**F**), PD-1int Tregs (**G**), and PD-1low Tregs (**H**) of total CD4+ T cells in the dLNs. Data are representative of n=3 independent experiments. Each symbol represents one mouse. Lines indicate means. One-way ANOVA with Bonferroni's multiple comparisons test.

FOXP3+ Tregs (*Figure 1B and F–H*). This data suggests that selected experimental scheme with the NucPrs promote rapid induction of Tfr in mice.

## NucPrs promote accumulation of Tfr with immunosuppressive gene expression profile

To assess gene expression and clonal repertoire of CXCR5high PD1high FOXP3+ Tfr induced by SA-NucPrs booster in mice, we performed 10× TCR immunorepertoire analysis of follicular (CXCR5high PD1high) T cells sorted from the dLNs of mice immunized with SA-DEL and boosted with SA or SA-NucPrs as described above (*Figure 1A*, *Figure 2*, *Figure 2—figure supplement 1A*). Graph-based clustering analysis of follicular T cells revealed 12 clusters of cells that express *Bcl6* with clusters 9–12 enriched

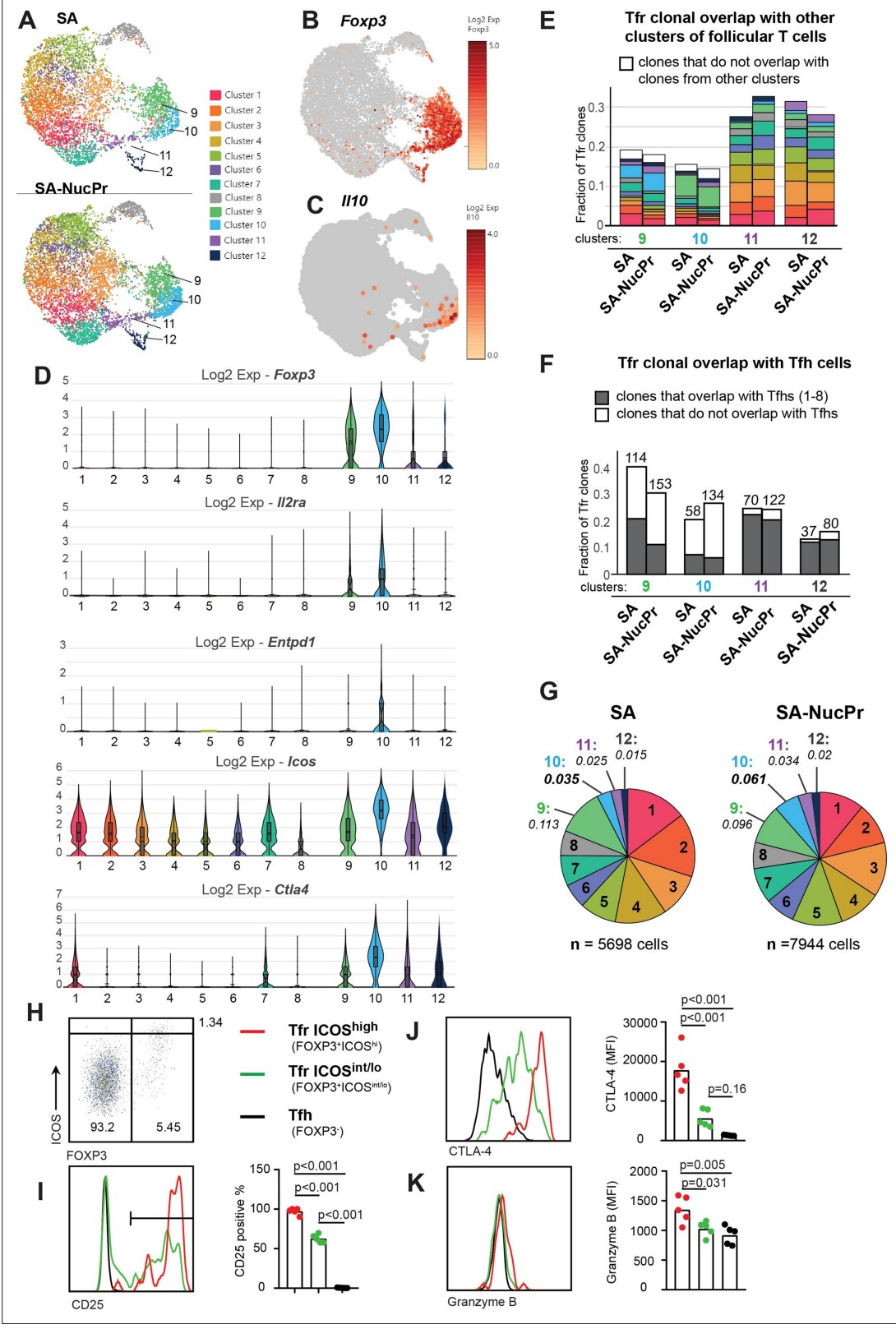

**Figure 2.** Analysis of follicular regulatory T cells (Tfr) gene expression and TCR repertoire in streptavidin (SA) and SA-nuclear protein (NucPr)-boosted mice (related to **Figure 2—figure supplement 1**). (**A–G**) Single-cell 10× Genomics (Cell Ranger) analysis of gene expression and TCR repertoire of the CD4+CD8-220-CXCR5hiPD1hi cells (Tfr and Tfh) sorted from the draining inguinal lymph nodes (dLNs) of mice immunized with SA-DEL and boosted with SA (two mice) or SA-NucPr (two mice) as in **Figure 2—figure supplement 1A**. (**A**) Graph-based clustering of follicular T cells from SA and SA-

*Figure 2 continued on next page*

*Figure 2 continued*

NucPr-boosted mice visualized in 2D using uniform manifold approximation and projection for dimension reduction (UMAP) algorithm. (**B, C**) Single-cell expression of *Foxp3* (in **B**) and *Il10* (in **C**) in follicular T cells for combined SA and SA-NucPr data shown in UMAP. (**D**) Expression of selected genes associated with Treg-mediated regulation in the follicular T cell clusters 1–12. Clusters 9–12 are enriched for follicular T cells with *Foxp3* expression and will be called Tfr-like cell clusters. Clusters 1–8 will be called Tfh-like cell clusters. (**E**) Overlap of TCR clones (with >1 cell per clone) within Tfr-like clusters 9–12 with other follicular T cell clusters (color-coded as in 2A). Note that the largest fraction of TCR clones in cluster 9 overlap with cluster 10 and vice versa. (**F**) Fraction of TCR clones (with >1 cell per clone) within Tfr-like clusters 9–12 that overlap or not with Tfh-like clusters 1–8 after SA versus SA-NucPr boosting. Note that there is no increase in the Tfh-like TCR clones within Tfr-like clones after SA-NucPr boosting. (**G**) Relative abundance of follicular T cells in 1–12 clusters in the SA versus SA-NucPr-boosted mice. Note about twofold increase in the Tfr-like cluster 10 in the SA-NucPr-boosted mice. (**H–K**) Flow cytometry analysis of CD25, CTLA4, and granzyme B expression in CD4$^+$CD8$^-$B220$^-$CXCR5$^{high}$PD1$^{high}$ cells that are FOXP3$^-$ (Tfh), FOXP3$^+$ICOS$^{high}$ (Tfr ICOS$^{high}$) and FOXP3 ICOS$^{int/low}$ (Tfr ICOS$^{int/lo}$) in the dLNs of SA-NucPr-boosted mice. (**H**) The gating strategy of CD4$^+$CD8$^-$B220$^-$CXCR5$^{high}$PD1$^{high}$ cells. (**I–K**) Representative flow histograms (left panels) and percent of positive cells (right panel) for CD25 (in **I**) or MFIs (right panels) for CTLA4 (in **J**), granzyme B (in **K**). n=2 independent experiments. Each point represents one mouse. One-way ANOVA with Bonferroni's multiple comparisons test.

The online version of this article includes the following source data and figure supplement(s) for figure 2:

**Figure supplement 1.** Analysis of follicular T cells gene expression in the streptavidin (SA) and SA-nuclear protein (NucPr)-boosted mice.

**Figure supplement 1—source data 1.** The genes differentially expressed in Tfh cells in the streptavidin (SA) versus SA-nuclear protein (NucPr)-boosted mice with statistically significant differences in expression.

**Figure supplement 2.** Analysis of human follicular T cells gene expression.

for expression of *Foxp3* (***Figure 2A–D***, ***Figure 2—figure supplement 1A–E***, ***Supplementary file 1***). Based on that and uniform manifold approximation and projection (UMAP) dimension reduction of the data, we will call clusters 9–12 as Tfr-like cells and clusters 1–8 as Tfh-like cells. Clusters 9 and 10 have majority of Tfr-like cells, while the cells in cluster 12 have upregulated expression of *Mki67*, *Ccna2,* and other genes associated with cell proliferation (***Figure 2A and B***, ***Figure 2—figure supplement 1D, E***, ***Supplementary file 1***). As previously reported (***Maceiras et al., 2017***), we found that TCR clonal repertoire of Tfr-like cells is very diverse, with majority of cells present at 1 cell per clone in the 10× data. We therefore performed analysis of the more abundant clones (>1 cell per clone) in ***Figure 2E and F***. Of note, the majority of TCR clones in clusters 9–12 overlapped with TCR clones from other follicular T cell clusters (***Figure 2E***). In clusters 9 and 10 at least half of the clones were found exclusively in other Tfr-like clusters. In contrast, in clusters 11 and 12 the majority of clones were also present in Tfh-like clusters (***Figure 2E and F***).

In parallel to the observed accumulation of CXCR5$^{high}$ PD1$^{high}$ FOXP3$^+$ cells in mice boosted with SA-NucPrs by flow cytometry (***Figure 1C and F***), 10× analysis revealed increase of cells in cluster 10 (***Figure 2A and G***) that had the highest expression of *Foxp3*, *Icos*, *Il2ra*, *Entpd1*, and *Ctla4* (***Figure 2D***). Cluster 10 was also enriched for cells expressing *Il10* and *granzyme B*, while *Nrn1* (encoding neuritin 1)- expressing cells were present in both clusters 9 and 10 and to some extent in other Tfr-like and Tfh-like cell clusters (***Figure 2C***, ***Figure 2—figure supplement 1F***). In accord with the 10× data for cluster 10, flow cytometry analysis confirmed that CXCR5$^{high}$ PD1$^{high}$ FOXP3$^+$ cells with the highest levels of ICOS were CD25$^{pos}$ and had elevated levels of CTLA4, and granzyme B, as compared to Tfr with ICOS$^{int/low}$ and Tfh cells (***Figure 2H–K***). Overall, the data suggests that the accumulating Tfr in SA-NucPr-boosted mice are likely to have more immunosuppressive Treg phenotype.

Importantly, we detected no increase in the Tfh-like cell TCR clones (from clusters 1–8) within Tfr-like clusters (9–12) in mice boosted with SA-NucPrs as compared to SA. Moreover, there was a trend for decreased frequency of Tfh-like cluster-associated clones among Tfr in mice boosted with SA-NucPr (***Figure 2E and F***). Therefore, based on the 10× TCR immunorepertoire analysis, we suggest that the observed increase in Tfr in the SA-NucPrs-boosted mice occurs due to accumulation of the cells with immunosuppressive phenotype that are not derived from abundant clones of Tfh cells.

Interestingly, single-cell analysis of Tfh cells (clusters 1–8) revealed a few genes with statistically different levels of expression in SA compared to SA-NucPrs-boosted mice (***Figure 2—figure supplement 1G***). For example, expression of proton-coupled peptide transporter *Slc15a2* (that was detected in a few Tfh cells) was completely abolished in SA-NucPrs-boosted mice. In addition, the gene *Apoe* encoding apolipoprotein E (expressed in Tfh cluster 1) was downregulated in the Tfr-induced mice (***Figure 2—figure supplement 1H***).

## Tfr-like cells with immunosuppressive gene expression profile in human lymph nodes

To determine whether human Tfr have any subsets similar to the murine Tfr with immunosuppressive phenotype, we performed PCA of the publicly available integrated multimodal single-cell data from human LN samples (GSE195673). We performed graph-based clustering analysis of CD4 T cells and identified follicular T cell clusters using a modular score generated based on the expression of *CXCR5*, *PDCD1*, *BCL6*, *ICOS*, *CTLA4*, *IL1R2*, and *CXCL13* (*Figure 2—figure supplement 2A*). Follicular T cell subsets were then reclustered and expression of *FOXP3* and other genes associated with Treg's immunosuppressive activity were analyzed (*Figure 2—figure supplement 2A-C*). Within eight of the detected follicular CD4 T cell clusters, FoxP3 expression was detected in clusters 4–7, with the highest expression in cluster 5 and the lowest in cluster 7. Elevated expression of *IL2RA* and *ENTPD1*, *CTLA4*, *LAG3* was detected in clusters 5–7. Expression of *TGFβ-1* and *ICOS* was the highest in cluster 7. Importantly, clusters 6 and 7 were also enriched in IL10-expressing cells. However, these clusters also expressed some Tfh-specific genes, such as *IL21* and *CD40L* (especially in cluster 6) (*Figure 2—figure supplement 2B, C*). Therefore, based on the single-cell analysis, human LNs have follicular T cells that resemble some gene expression characteristics of the immunosuppressive murine Tfr subset (cluster 10), but have reduced or undetectable expression of *FOXP3* and express some factors typical for Tfh cells.

## Tfr specifically suppress expansion of GC B cells that acquire NucPrs and exert more modest inhibition of total GC B cells

To determine whether NucPrs-induced Tfr can in turn affect GC responses, we examined the total and SA-specific GC B cells in the SA-DEL-immunized mice boosted with SA-NucPrs or control Ags (*Figure 1A*). We found that in mice boosted with SA-NucPrs the frequency of GC B cells and the size of GCs were reduced compared to mice boosted with SA (*Figure 3A and B*, *Figure 3—figure supplement 1A, B*). Of note, while SA-boosting induced about threefold increase in the GC response as compared to no boosting control, no significant increase in GCs was detected in the SA-NucPrs-boosted mice (*Figure 3—figure supplement 1C*). Moreover, SA-specific GC B cells that were expected to preferentially reacquire SA-linked NucPrs underwent greater suppression than total GC B cells or DEL-specific B cells (*Figure 3A, C and D*; *Figure 3—figure supplement 1D–J*). Importantly, in mice boosted with SA-OVA no decrease in the total or SA-specific GC B cells was detected (*Figure 3A–D*). In parallel to the observed reduction in the numbers of SA-specific GC B cells, we detected reduced accumulation of the SA-specific class-switched memory B cells in mice boosted with SA-NucPrs (*Figure 3E–G*). We also found decreased formation of SA-specific plasmablasts (PB), including the PB recently derived from GC B cells (GCPB) (*Figure 3H–L*). Finally, in mice boosted with SA-NucPrs the titers of the SA-specific Abs were reduced as compared to mice boosted with SA (*Figure 3—figure supplement 1K*).

Importantly, our suggested immunization scheme (*Figure 1A*) promoted suppression of the NucPrs-acquiring GC B cells not only in the B6 mice but also in NZM2328 mice. NZM2328 mice spontaneously develop dsDNA auto-Abs and systemic lupus erythematosus-like disease after 5 months of age (*Rudofsky and Lawrence, 1999*; *Waters et al., 2001*; *Wolf et al., 2018*) and form anti-ribonucleoprotein (RNP) auto-Abs (*Figure 4A*). In this study we found that the ratio of GC B cells to Tfh cells (that is highly conserved in GCs; *Baumjohann et al., 2013*) is elevated over fivefold in 6- to 8-week-old female NZM2328 compared to B6 mice, suggesting dysregulated GC responses even in young NZM2328 mice (*Figure 4B and C*). Importantly, SA-DEL immunization followed by boosting with SA-NucPrs decreased the GC B/Tfh cell ratio in the NZM2328 mice, as well as in B6 mice (*Figure 4B and C*). Even more significant was the observed decrease in the SA-specific GC B cell/Tfh cell ratio in the SA-NucPrs-boosted NZM2328 (*Figure 4D and E*). The observed suppression of the GC responses was likely due to increased Tfr responses in the SA-NucPrs-boosted mice, although in different sets of NZM2328 mice Tfr levels were variable (*Figure 4F*). Therefore, even in pre-autoimmune state in mice where B cell responses are inflated, SA-DEL immunization followed by SA-NucPrs boosting provided a suppressive effect.

To verify whether observed suppression of the GC responses in the SA-NucPrs-boosted mice depended on Tfr, we repeated the immunization/boosting experiments (as in *Figure 1A*) in *Foxp3*[cre] *Bcl6*[fl/fl] mice that are deficient for Tfr (*Wu et al., 2016*) or in control *Foxp3*[cre] *Bcl6*[+/+] mice. As expected,

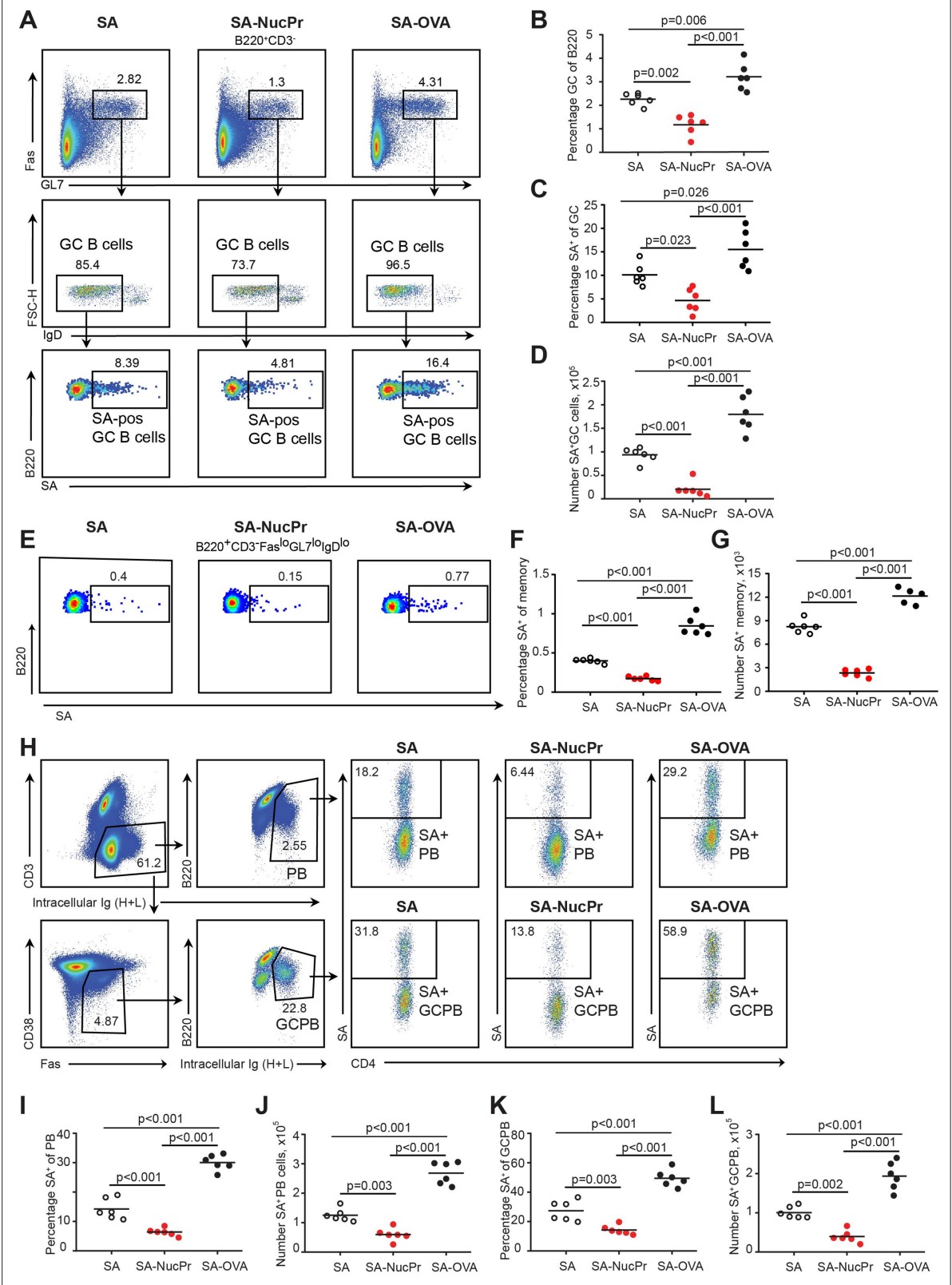

**Figure 3.** Boosting mice with streptavidin nuclear proteins (SA-NucPrs) suppresses germinal center (GC), memory and plasmablasts (PB) responses with predominant inhibition of the SA-specific B cells (related to *Figure 3—figure supplement 1*). Flow cytometry analysis of the total and SA-specific GC (A–D), memory (E–G), and PB (H–L) responses in the draining inguinal lymph nodes (dLNs) of mice treated as shown in *Figure 1A*. (A) Representative flow plots for the total and SA-specific GC B cells in SA, SA-NucPr, and SA-OVA-boosted mice. (B) The GC B cells percentage of total B220[+] B cells.

*Figure 3 continued on next page*

*Figure 3 continued*

(**C, D**) SA-specific GC B cells percentage of total GC B cells (**C**) and their numbers in dLNs (**D**). (**E**) Representative flow plots for the class-switched SA-specific memory B cell response in SA, SA-NucPr, and SA-OVA-boosted mice. (**F, G**) SA-specific memory B cells percentage of total class-switched memory B cells (B220[+]CD3[-]FAS[lo]GL7[lo]IgD[lo]) (**F**) and their numbers in dLNs (**G**). (**H**) Representative flow plots for SA-specific PB and GC B cell (GCPB) response in SA, SA-NucPr, and SA-OVA-boosted mice. (**I–L**) SA-specific PB and GCPB percentage of total PB and GCPB (**I, K**) and their numbers in dLNs (**J, L**). Data are representative of n=3 independent experiments. Each symbol represents one mouse. Lines indicate means. One-way ANOVA with Bonferroni's multiple comparisons test.

The online version of this article includes the following figure supplement(s) for figure 3:

**Figure supplement 1.** Boosting mice with streptavidin nuclear proteins (SA-NucPrs) suppresses germinal center (GC) and antibody (Ab) responses with predominant inhibition of the SA-specific B cells (related to *Figure 3*).

the frequency of CXCR5[high] PD1[high] FoxP3[+] Tfr in the *Foxp3[cre] Bcl6[fl/fl]* mice was low, and no increase in Tfr was observed after boosting these mice with SA-NucPrs (as opposed to *Foxp3[cre] Bcl6[+/+]* control mice) (*Figure 5A–C*). At the same time, Tfh responses in *Foxp3[cre] Bcl6[fl/fl]* mice were not significantly affected as compared to *Foxp3[cre] Bcl6[+/+]* controls (*Figure 5D and E*). Importantly, while boosting *Foxp3[cre] Bcl6[+/+]* mice with SA-NucPrs promoted significant suppression of the SA-specific and total GC B cells, both effects were completely abolished in the Tfr-deficient *Foxp3[cre] Bcl6[fl/fl]* mice (*Figure 5F–I*). This data suggests a direct role of NucPrs-induced Tfr in the negative regulation of the GC responses with predominant suppression of the NucPrs-acquiring GC B cells.

## Identification of the immunization strategy that triggers rapid accumulation of Tfr and suppression of GC response

We then examined which immunization strategy promotes accumulation of the NucPrs-induced Tfr (*Figure 6*). Direct immunization of naïve mice with SA-NucPrs did not induce rapid expansion of Tfr that was observed in mice preimmunized with SA-DEL (*Figure 6A–D*, left panels). We then assessed whether anti-SA Abs (that should be induced after preimmunization of mice with SA-DEL) may facilitate SA-NucPrs uptake, presentation and efficient activation of Tfr by antigen-presenting cells. To address this, naïve mice were transferred with serum from SA-immunized mice and then immunized with SA-NucPrs. However, at 3 days following vaccination no effect on the Tfr frequency was detected as compared to mice that did not get serum transfer (*Figure 6A, C and E*, left panels). We next examined whether accumulation of SA-specific B cells induced by preimmunization of mice with SA-DEL was important for rapid increase in Tfr. To test this, we compared induction of the Tfr response by SA-NucPrs boosting in mice preimmunized with SA-DEL or with OVA. In contrast to the rapid expansion of Tfr and suppression of GCs in the SA-DEL-preimmunized mice, no increase in Tfr and/or suppression of GC B cells was detected in mice preimmunized with OVA (*Figure 6B, D and F*). Since boosting of mice with an ongoing SA-specific GC response with SA-NucPrs is expected to promote acquisition of these Ags by the SA-specific GC B cells, we suggest that directing a combination of NucPrs to a selected subset of GC B cells may be a quick and efficient method for triggering rapid induction of Tfr and suppression of GC B cell subset that present the acquired NucPrs' peptides.

We then examined which of the selected NucPr Ags (nucleosomes, SSA-Ro, RNP-Sm, Scl70, and Jo1) promotes the observed accumulation of Tfr and suppression of GC B cells. To address this, SA-DEL-preimmunized mice were boosted with SA conjugated to each one of these Ags to assess Tfr frequency and SA-specific GC response (*Figure 6—figure supplement 1A–D*). While robust increase in Tfr was detected in mice boosted with combined SA-NucPrs (*Figure 6—figure supplement 1B*), boosting with separate SA-NucPr Ags did not promote significant increase in Tfr (*Figure 6—figure supplement 1C*). Based on this data we suggest that the observed increase in Tfr (*Figure 6—figure supplement 1B*) occurs due to additive accumulation of Tfr specific to various NucPrs Ags.

Despite the non-significant increase in Tfr after targeting separate NucPrs to SA-specific B cells, we detected decreased frequency of the SA-specific GCs B cells in mice boosted with all selected Ags except SA-Jo1 (*Figure 6—figure supplement 1D*). In addition, we have tested the frequency of the nucleosome-specific B cells in mice boosted with SA-nucleosomes conjugates. In accord with previous study, we found (based on the staining with fluorescent nucleosome tetramers) that about ~1% of GC B cells in SA-DEL-immunized mice boosted with SA were specific to nucleosomes (*Gonzalez-Figueroa et al., 2021*). However, boosting of mice with SA-nucleosomes led to significant drop in the frequency and numbers of the nucleosome-specific GC and memory cells (*Figure 6—figure supplement 1E–H*).

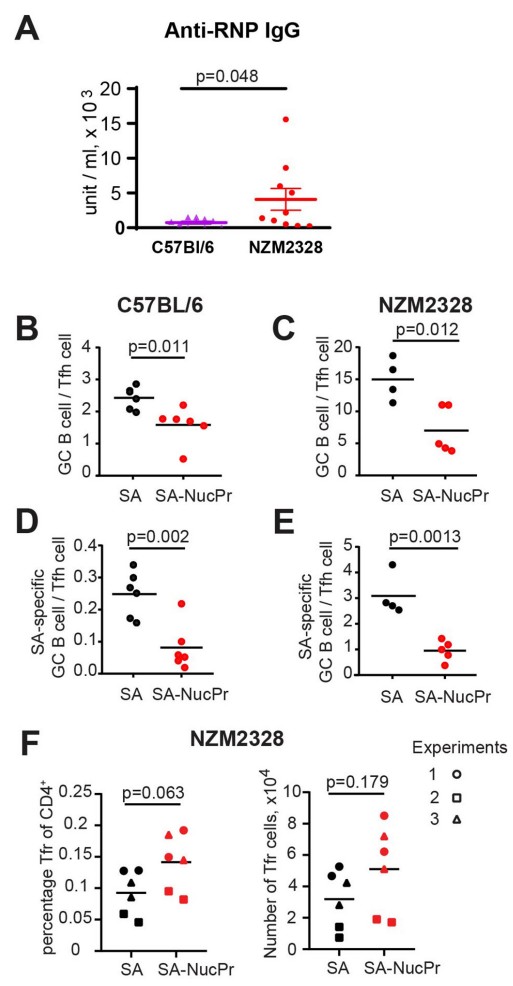

**Figure 4.** Boosting mice with streptavidin nuclear proteins (SA-NucPrs) reduces the ratio of germinal center (GC) B cells to Tfh cells in both C57BL/6 and NZM2328 mice. (**A**) Analysis of anti-ribonucleoprotein (RNP) antibodies (Abs) via ELISA in the serum of female C57BL/6 (52 weeks of age) and NZM2328 female mice (32–52 weeks of age). (**B–F**) Suppression of GC B cell responses and induction of follicular regulatory T cell (Tfr) response in the draining inguinal lymph nodes (dLNs) of 6- to 8-week-old C57BL/6 and NZM2328 female mice treated as in *Figure 1A* after the boost with SA-NucPr. (**B–E**) The ratio of GC B cells to Tfh cells in C57BL/6 (**B**) and NZM2328 (**C**) mice. n=2 independent experiments. The ratio of SA-specific GC B cells to Tfh cell in C57BL/6 (**D**) and NZM2328 (**E**) mice. (**F**) Tfr percentage of CD4+ cells (left panel) and total numbers (right panel) in the NZM mice. Different symbols represent n=3 independent experiments. Each symbol represents one mouse. Lines indicate means. Two-tailed Student's t test.

Overall, this data suggests that nucleosomes and other NucPr Ags contain peptides cognate to Tfr and if acquired by GC B cells may promote their suppression by Tfr.

## Formation of conjugates between human B cells acquiring NucPrs and Tfr

While antigen-specific targeting of NucPrs to GC B cells promotes rapid accumulation of Tfr in mice, whether this finding maybe relevant to the regulation in humans is unclear. We therefore assessed whether human Tfr may have a subset of cells cognate to the NucPrs. Based on the previous studies, cognate interactions between B and T cells cocultured ex vivo lead to increased formation of B-T cell conjugates detectable by flow cytometry analysis (*Choudhuri et al., 2005*). We therefore utilized a similar approach with Tfr (see *Figure 7A*) and B cells (CD19+ CD3- CD27-) sorted from the blood of healthy human subjects. To promote antigen uptake by human B cells we first coupled anti-human IgM-biotin Abs to SA, SA-DEL or SA-NucPrs to generate protein complexes αIgM (αIgM-SA or αIgM-SA-DEL) and αIgM-NucPrs (αIgM-SA-NucPrs). Sorted B cells were then cultured in the presence of αIgM or αIgM-NucPrs to promote crosslinking of B cell receptors and internalization of the linked to αIgM protein complexes for degradation and HLA class II /peptide presentation. Tfr from the same patient were then cocultured with activated or control B cells for 36 hr and formation of Tfr-B cell conjugates was assessed by flow cytometry analysis (*Figure 7B*). In three independent experiments we detected an increase in the frequency of Tfr-B cell conjugates when Tfr were cocultured with activated B cells that have internalized NucPrs as compared to control Ags or B cells that did not acquire Ags (*Figure 7B and C*). Based on that, we suggest that a significant fraction of the circulating human Tfr may be specific to the peptides from the NucPrs selected in this study.

## Discussion

In this study we showed that combination of NucPrs (SS-A/Ro, RNP-Sm, Scl70, Jo-1, and nucleosomes) selected based on their frequent targeting by auto-Abs in autoimmune diseases, induces rapid accumulation of Tfr in the dLNs. While there is no detectable Tfr increase in mice directly immunized with NucPrs (conjugated to SA and administered in Ribi adjuvant), a rapid Tfr response is detected when SA-NucPrs are administered to mice with an ongoing SA-specific GC response. In these mice, SA-specific GC (and possibly

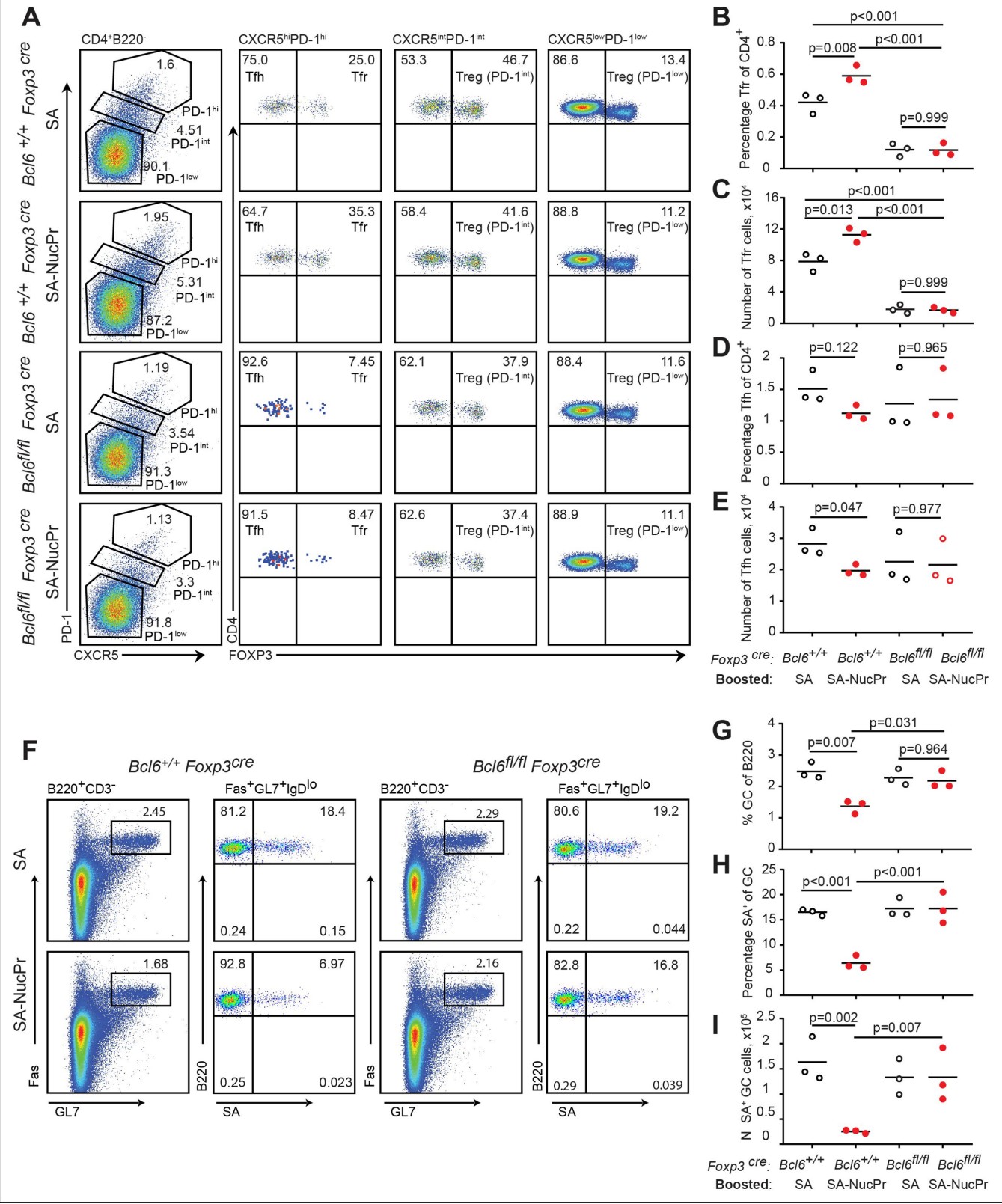

**Figure 5.** Follicular regulatory T cells (Tfr) are required for the suppression of germinal center (GC) response in mice boosted with streptavidin nuclear proteins (SA-NucPrs). *Bcl6+/+ Foxp3* Cre and Tfr-deficient *Bcl6fl/fl Foxp3Cre* mice were subcutaneously (s.c.) immunized with SA-DEL in Ribi and at day 8 reimmunized with SA, or SA-NucPr in Ribi s.c. for analysis 3 days later. (**A**) Representative flow plots for Tfh, Tfr, and other Treg subsets in the dLNs of mice boosted with SA or SA-NucPr. (**B, C**) Tfr percentage of CD4+ T cells (**B**) and total numbers (**C**) in dLNs. (**D, E**) Tfh cells percentage of CD4+ T cells

*Figure 5 continued*

(**D**) and total numbers (**E**) in dLNs. (**F**) Representative flow plots showing GC B cells and SA-specific GC B cells in Tfr-proficient and -deficient mice after the boost with SA or SA-NucPr. (**G**) GC B cells percentage of total B220⁺ B cells. (**H, I**) SA-specific GC B cells percentage of total GC B cells (**H**) and total numbers (**I**) in dLNs. n=3 independent experiments. Each symbol represents one mouse. Lines indicate means. Two-way ANOVA with Tukey's multiple comparisons test.

memory) B cells are expected to acquire SA-NucPrs via their B cell receptors, internalize the Ags, and present both antigenic and NucPrs peptides in complex with MHCII for recognition by follicular T cells. While our study does not rule out the potential for dendritic cells and/or other Ag-presenting cells to promote Tfr responses at longer time scale, it suggests that direct targeting of selected NucPrs to abundant antigen-specific B cells via BCR could be utilized to trigger very rapid and robust increase in Tfr. Interestingly, a previous study suggested pausing of Tfr in proximity of the tingible body macrophages that acquire dying B cells within GCs (*Jacobsen et al., 2021*). It is plausible to suggest that these macrophages may be another cell type in GCs that presents NucPr peptides for cognate stimulation of Tfr, possibly under all immunization conditions.

Based on the 10× Genomics gene expression analysis, the majority of the NucPrs-induced Tfr have distinct gene expression characteristics that suggest enhanced immunosuppressive functions. Based on the TCR immunorepertoire analysis, Tfr subset enriched in the SA-NucPrs-boosted mice (cluster 10) have significant clonal overlap with other Tfr subsets. In contrast to this 'immunosuppressive' subset, Tfr-like cell clusters 11 and 12 are predominantly overlapping with Tfh-like cell clones and may be related to the recently identified subset of Tfh cells in GCs that upregulate FOXP3 and play a role in the shutoff of GC response over time (*Jacobsen et al., 2021*). Importantly, in mice boosted with SA-NucPrs we detected no increase in the Tfh-related clones in the 'immunosuppressive', as well as all

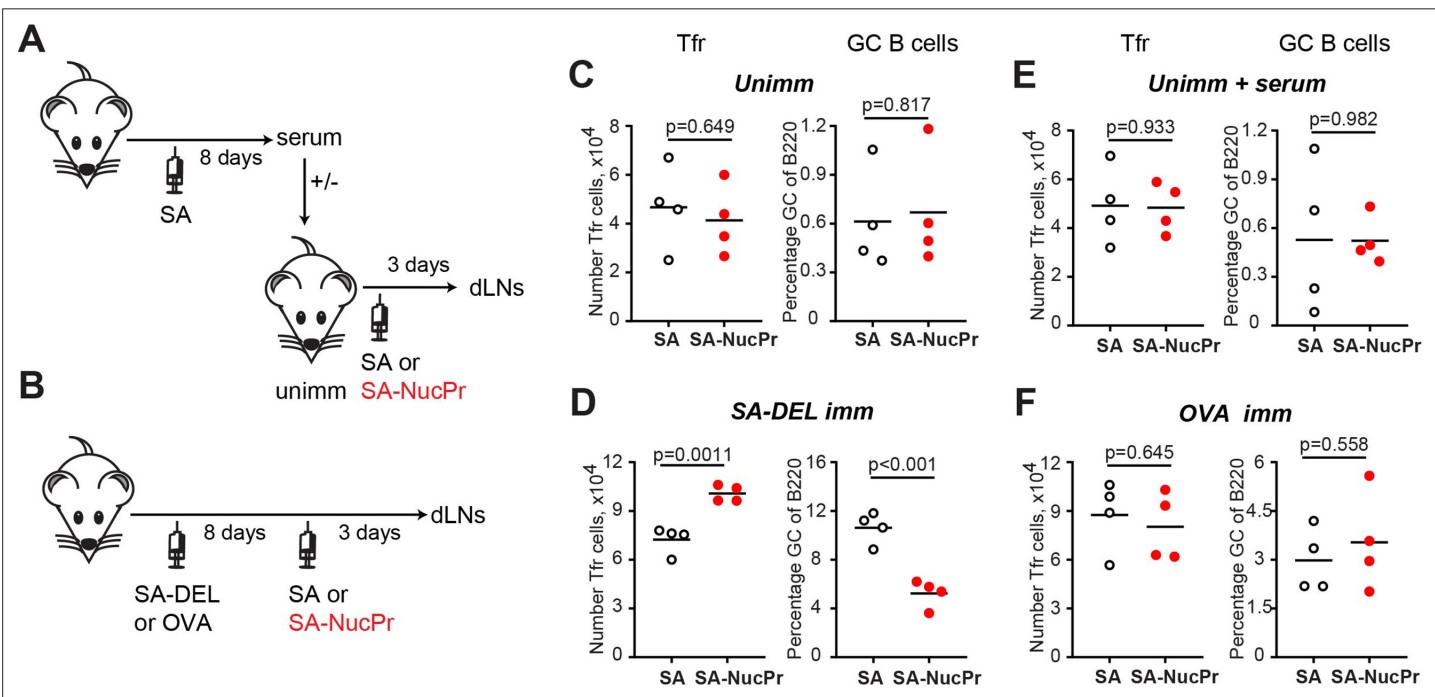

**Figure 6.** Analysis of the immunization conditions inducing follicular regulatory T cell (Tfr) response. (**A, C, E**) Experimental outline (**A**). Some B6 mice were subcutaneously (s.c.) immunized with streptavidin (SA) and serum were collected at day 8. Unimmunized B6 mice were transferred with serum from above (**E**) or not (**C**) and were s.c. immunized with SA or SA-nuclear proteins (NucPr) in Ribi for analysis 3 days later. (**B, D, F**) Experimental outline (**B**). B6 mice were s.c. immunized with SA-DEL (**D**) or OVA (**F**) in Ribi and at day 8 were s.c. boosted with SA or SA-NucPr in Ribi for analysis 3 days later. (**C, D, E, F**) The numbers of Tfr (left panels) and the GC B cells percentages of total B220⁺ B cells (right panels). Data are from n=2 independent experiments. Each symbol represents one mouse. Lines indicate means. Two-tailed Student's t test.

The online version of this article includes the following figure supplement(s) for figure 6:

**Figure supplement 1.** Analysis of follicular regulatory T (Tfr) and germinal center (GC) B cell responses in mice boosted with streptavidin (SA) linked to individual nuclear proteins (NucPrs)/complexes.

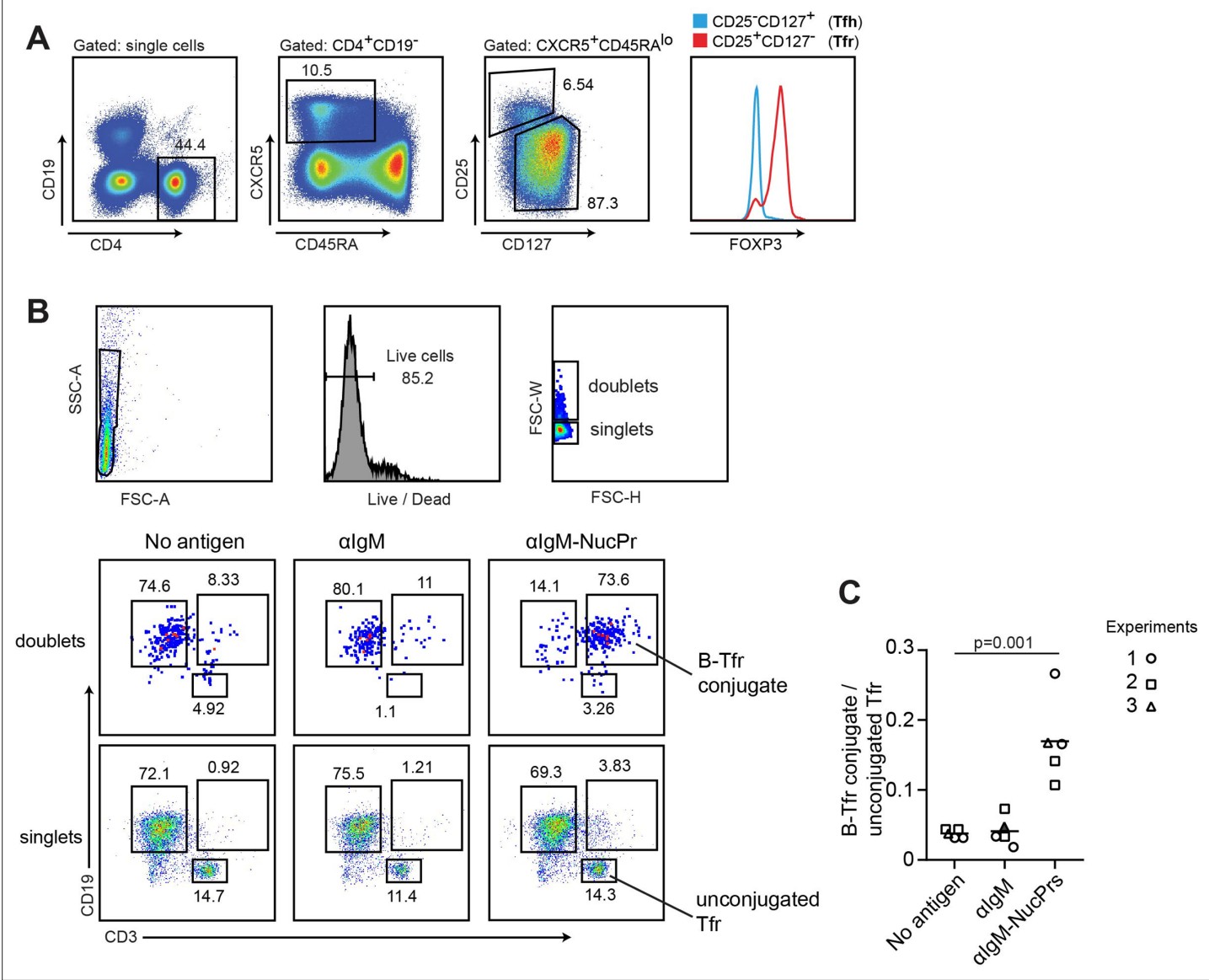

**Figure 7.** Formation of conjugates between human B cells acquiring nuclear proteins (NucPrs) and follicular regulatory T cells (Tfr). B cells and Tfr were sorted from human blood. (**A**) Sorting strategy for Tfr. (**B, C**) Sorted CD19$^+$CD3$^-$CD27$^-$ naïve B cells were incubated ex vivo with αIgM (anti-IgM-SA [opened symbols], anti-IgM-SA-DEL [filled symbol]), αIgM-NucPrs (anti-IgM-SA-NucPrs), or medium for control for 30 min. After that they were cocultured with Tfr from the same patient. (**B**) Representative example of flow cytometry analysis of singlets and doublets for B-Tfr conjugate formation after 36 hr of coculturing. (**C**) The ratio of Tfr-B cell conjugates (in doublets) to unbound Tfr (in singlets). Data are from n=3 independent experiments. Each type of symbol represents one patient, one or two replicas. Two-tailed Student's t test.

other Tfr-like subsets. Therefore, we suggest that the majority of NucPrs-induced Tfr are not derived from Tfh cells. Based on this, on the previously established overlap of Tfr and Treg TCR repertoire, as well as the important role of Ag-specific B cells in the very rapid induction (3 days) of the immunosuppressive Tfr, we speculate that majority of NucPrs-induced Tfr accumulate due to proliferation of preexisting Tfr clones in the follicles or their precursors at the T/B border following their cognate interactions with NucPr-acquiring and -presenting B cells.

Identification of the dominant self-Ags cognate to Tregs remains an important task for both fundamental analysis of thymus-derived Tregs and development of Tregs-dependent translational therapies (*Leonard et al., 2017*). Previous studies suggested that histones may contain Tregs epitopes. Vaccine therapy with peptides from nucleosomal histones H4 and H1 promoted Treg response, diminished auto-Ab levels, and delayed the onset of nephritis in lupus-prone mice (*Kang et al., 2011*; *Kang*

*et al., 2005*). However, availability of Tregs epitopes in the other NucPr Ags has not been suggested before. Because we detected the strongest Tfr response after administration of the combination of NucPrs, we suggest that selected self-Ags contain multiple protein epitopes cognate to Tfr. Importantly, our initial analysis of Tfr isolated from human blood suggests that a fraction of these cells may be specific to the peptides from the NucPrs that we utilized in the study. Future studies should carefully delineate the peptide epitopes within NucPrs that induce Tfr responses in mice and humans and assess their potential for future vaccine therapies.

Our data suggest that Tfr induced by SA-NucPrs boosting promote partial suppression of the overall GC response. Importantly, they more strongly impede expansion of the SA-NucPrs-acquiring GC B cells and reduce their memory and Ab-secreting cell responses. Based on that, we conclude that cognate interactions with Tfr promote superior suppression of GC B cells. The critical role of Tfr in the observed regulation (rather than indirect effects from SA-NucPrs administration) is supported by complete reversal of the observed immunosuppression in the Tfr-deficient mice. To summarize, while our study does not rule out non-specific inhibitory effect of Tfr on non-cognate GC B cells, it suggests that GC B cells that acquire selected NucPrs will be subjected to specific repression by cognate Tfr.

Multiple molecular mechanisms could potentially contribute to the negative control of GC B cells by cognate and non-cognate Tfr. NucPrs-induced Tfr upregulate expression of multiple genes that may potentially contribute to the negative regulation of B cells and Tfh cells, including *Ctla4*, *Icos*, *Cd39*, *granzyme B,* and *Il10*. Of note, expression of *neuritine* that was recently shown to be produced by Tfr and to limit B cell differentiation into plasma cells (*Gonzalez-Figueroa et al., 2021*) was not limited to the accumulating Tfr cluster with immunosuppressive phenotype. We speculate that multiple molecular mechanism may work in parallel to promote suppression of GC B cells. Cognate recognition by Tfr of the GC-presented MHCII/peptides likely leads to more prolonged interactions between the cells (*Jacobsen et al., 2021*), increasing the duration of inhibitory signals received by GC B cells from cognate Tfr. Future studies should dissect the contribution of distinct molecular pathways to the negative signals provided to GC B cells by Tfr.

Future studies should also examine potential engagement of the NucPrs-specific Tfr for translational applications. To this end, our study suggests that SA-NucPrs boosting promotes a significant suppression of the NucPrs-acquiring GC B cells not only in the wild-type (WT), but also in the lupus-prone NZM2328 mice that have dysregulated GC responses and over time develop anti-nuclear Abs. We therefore suggest that targeting NucPrs (and/or their specific peptides) to GC B cells should be further explored for analysis of their therapeutic potential for control of pathogenic B cells specific to autoantigens or/and allergens or promoting autoepitopes spreading for activation of pathogenic autoreactive T cells.

# Methods

**Key resources table**

| Reagent type (species) or resource | Designation | Source or reference | Identifiers | Additional information |
|---|---|---|---|---|
| Antibody | Anti-mouse B220 Biotin, clone RA3-6B2 (Rat monoclonal) | BD Bioscience | Cat#553086; PRID:AB 394616 | FC (1:100) |
| Antibody | Anti-mouse B220 Pacific Blue, clone RA3-6B2 (Rat monoclonal) | BioLegend | Cat#103227; PRID:AB 492865 | FC (1:100) |
| Antibody | Anti-mouse B220 PerCP-Cy5.5, clone RA3-6B2 (Rat monoclonal) | BD Bioscience | Cat#552771; PRID:AB 394457 | FC (1:100) |
| Antibody | Anti-mouse B220 V500, clone RA3-6B2 (Rat monoclonal) | BD Bioscience | Cat#561226; PRID:AB 10563910 | FC (1:100) |
| Antibody | Anti-mouse Bcl6 Alexa Fluor 488, clone K112-91 (Mouse monoclonal) | BD Bioscience | Cat#561524; PRID:AB 10716202 | IC (1:20) |

*Continued on next page*

*Continued*

| Reagent type (species) or resource | Designation | Source or reference | Identifiers | Additional information |
|---|---|---|---|---|
| Antibody | Anti-mouse CD3 Alexa Fluor 700, clone 17A2 (Rat monoclonal) | BioLegend | Cat#100216; PRID:AB 493697 | FC (1:100) |
| Antibody | Anti-mouse CD3 PE-CF594, clone 145–2C11 (Hamster monoclonal) | BD Bioscience | Cat#562286; PRID:AB 11153307 | IC (1:100) |
| Antibody | Anti-mouse CD4 APC/Cy7, clone RM4-5 (Rat monoclonal) | BioLegend | Cat#100526; PRID:AB 312727 | FC (1:100) |
| Antibody | Anti-mouse CD4 FITC clone RM4-5 (Rat monoclonal) | BD Bioscience | Cat#553047; PRID:AB 394583 | FC (1:100) |
| Antibody | Anti-mouse CD4 V500 clone RM4-5 (Rat monoclonal) | BD Bioscience | Cat#560782; PRID:AB 1937315 | FC (1:100) |
| Antibody | Anti-mouse CD8a APC/Cy7, clone 53–6.7 (Rat monoclonal) | BioLegend | Cat#100714; PRID:AB 312753 | FC (1:100) |
| Antibody | Anti-mouse CD8a Pacific Blue, clone 53–6.7 (Rat monoclonal) | BioLegend | Cat#100725; PRID:AB 493425 | FC (1:100) |
| Antibody | Anti-mouse CD8a V500 clone 53–6.7 (Rat monoclonal) | BD Bioscience | Cat#560776; PRID:AB 1937317 | FC (1:100) |
| Antibody | Anti-mouse CD8a Biotin clone 53–6.7 (Rat monoclonal) | BD Bioscience | Cat#553029; PRID:AB 394567 | FC (1:100) |
| Antibody | Anti-mouse CD25 BV421, clone PC61 (Rat monoclonal) | BioLegend | Cat#102034; PRID:AB11203373 | FC (1:100) |
| Antibody | Anti-mouse CD38 PerCP-Cy5.5, clone 90/CD38 (Rat monoclonal) | BD Bioscience | Cat#562770; PRID:AB 2737782 | FC (1:100) |
| Antibody | Anti-mouse CD95 PE-Cy7, clone Jo2 (Armenian hamster Monoclonal) | BD Bioscience | Cat#557653; PRID:AB 396768 | FC (1:100) |
| Antibody | Anti-mouse CD185 (CXCR5) BV605, clone L138D7 (Rat monoclonal) | BioLegend | Cat#145513; PRID:AB 2562208 | FC (1:50) |
| Antibody | Anti-mouse CD279 (PD-1) PE-Cy7, clone RMP1-30 (Rat monoclonal) | BioLegend | Cat#109110; PRID:AB 572017 | FC (1:100) |
| Antibody | Anti-mouse Foxp3 APC, clone FJK-16s (Rat monoclonal) | eBioscience | Cat#17-5773-82; PRID:AB 469457 | FC (1:100) |
| Antibody | Anti-mouse GL7 eFluor 450, clone GL-7 (Rat monoclonal) | eBioscience | Cat#48-5902-82; PRID:AB 10870775 | FC (1:100) |
| Antibody | Anti-mouse GL7 eFluor 660, clone GL-7 (Rat monoclonal) | eBioscience | Cat#50-5902-82; PRID:AB 2574252 | FC (1:100) |
| Antibody | Anti-mouse IgD APC/Cy7 clone 11–26c.2a (Rat monoclonal) | BioLegend | Cat#405716; PRID:AB 10662544 | FC (1:100) |
| Antibody | Anti-mouse IgG(H+L) HRP (Goat polyclonal) | Thermo Fisher Scientific | Cat#62–6520; PRID:AB 88369 | ELISA (1:2000) |
| Antibody | Anti-mouse Ig(H+L) Alexa Fluor 488 (Goat polyclonal) | Southern Biotech | Cat#1010–30; PRID:AB 2794130 | FC (1:100) |
| Antibody | Anti-human CD3 AF700, clone OKT3 (Mouse monoclonal) | BioLegend | Cat#317339; PRID:AB 2563407 | FC (1:100) |
| Antibody | Anti-human CD19 Pacific Blue, clone SJ25C1 (Mouse monoclonal) | BioLegend | Cat#363036; PRID:AB 2632787 | FC (1:100) |

*Continued on next page*

*Continued*

| Reagent type (species) or resource | Designation | Source or reference | Identifiers | Additional information |
|---|---|---|---|---|
| Antibody | Anti-human FoxP3 AF488, clone 259D (Mouse monoclonal) | BioLegend | Cat#320211; PRID:AB 430886 | FC (1:20) |
| Antibody | Anti-human CD4 BV510 clone RPA-T4 (Mouse monoclonal) | BioLegend | Cat#300545; PRID:AB 2563313 | FC (1:100) |
| Antibody | Anti-human CD127 BV605, clone A019D5 (Mouse monoclonal) | BioLegend | Cat#351333; PRID:AB 2562019 | FC (1:100) |
| Antibody | Anti-human CD25 PE, clone M-A251 (Mouse monoclonal) | BioLegend | Cat#356103; PRID:AB 2561860 | FC (1:100) |
| Antibody | Anti-human CD45RA APC-Cy7, clone HI100 (Mouse monoclonal) | BioLegend | Cat#304127; PRID:AB 10708419 | FC (1:100) |
| Antibody | Anti-human CXCR5 AF647, clone RF8B2 (Rat monoclonal) | BioLegend | Cat#558113; PRID:AB 2737606 | FC (1:50) |
| Antibody | Anti-human CD27 FITC clone M-T271 (Mouse monoclonal) | BD Bioscience | Cat#555440; PRID:AB 395833 | FC (1:100) |
| Antibody | Anti-human IgM Biotin F(ab')2 fragment (Goat polyclonal) | Jackson Immuno Research Labs | Cat#109-066-129; PRID:AB 2337642 | 5 µg/ml for activation |
| Chemical compound, drug | Ficoll-Paque Plus | GE Healthcare | Cat#17-1440-02 | |
| Commercial assay or kit | EZ-Link Sulfo-NHS-LC-Biotin | Thermo Fisher Scientific | Cat#A39257 | |
| Chemical compound, drug | Fluoromount-G | SouthernBiotech | Cat#0100–01 | |
| Commercial assay or kit | 1-step Ultra TMB-ELISA Substrate Solution | Thermo Fisher Scientific | Cat#34028 | |
| Commercial assay or kit | Foxp3/Transcription Factor staining Buffer Set | eBioscience | Cat#00-5523-00 | |
| Chemical compound, drug | Gelatin blocking buffer 1% in PBS | Alfa Aesar | Cat#J62755 | |
| Peptide, recombinant protein | Recombinant Human CD40L | BioLegend | Cat#591702 | |
| Peptide, recombinant protein | Ovalbumine | Sigma | Cat#A5503-5G | |
| Peptide, recombinant protein | Duck Egg Lysozyme | (Allen, C.D. et al., 2007). | | Purified from Duck eggs |
| Chemical compound, drug | CM Sephadex C-25 beads | GE Healthcare | Cat#17-0210-01 | |
| Chemical compound, drug | Sephadex G-50 medium beads | GE Healthcare | Cat#17-0043-01 | |
| Chemical compound, drug | Sephadex G-100 beads | GE Healthcare | Cat#17-0060-01 | |
| Chemical compound, drug | Sigma Adjuvant System | Sigma | Cat#S6322-1VL | |
| Peptide, recombinant protein | Streptavidin | Sigma | Cat#S4762-5MG | |
| Peptide, recombinant protein | Streptavidin PE | BioLegend | Cat#405204 | |
| Peptide, recombinant protein | Streptavidin Qdot 605 | Thermo Fisher Scientific | Cat#Q10101MP | |
| Peptide, recombinant protein | Streptavidin Qdot 647 | Thermo Fisher Scientific | Cat#S21374 | |
| Peptide, recombinant protein | Streptavidin Qdot 655 | Thermo Fisher Scientific | Cat#Q10121MP | |
| Peptide, recombinant protein | Nucleosome | AROTEC DIAGNOSTICS | ATN02-02 | |

*Continued on next page*

*Continued*

| Reagent type (species) or resource | Designation | Source or reference | Identifiers | Additional information |
|---|---|---|---|---|
| Peptide, recombinant protein | Jo-1 | AROTEC DIAGNOSTICS | ATJ01-02 | |
| Peptide, recombinant protein | Scl-70 | AROTEC DIAGNOSTICS | ATS01-02 | |
| Peptide, recombinant protein | Ro (SSA) | AROTEC DIAGNOSTICS | ATR02-02 | |
| Peptide, recombinant protein | RNP-Sm | AROTEC DIAGNOSTICS | ATR01-02 | |
| Strain, strain background (*Mus musculus*) | C57BL/6J | The Jackson Laboratory | JAX:000664 | Wild-type (males and females) |
| Strain, strain background (*Mus musculus*) | *Bcl6<sup>fl/fl</sup>* | *Hollister et al., 2013* | | (Males and females) |
| Strain, strain background (*Mus musculus*) | *Foxp3-Yfp<sup>cre</sup>* | *Rubtsov et al., 2008* | | (Males and females) |
| Strain, strain background (*Mus musculus*) | NZM2328 | *Jacob et al., 2003* | | Females |
| Software, algorithm | FlowJo V10 | FlowJo | https://www.flowjo.com/ | |
| Software, algorithm | GraphPad Prism 8 | GraphPad | https://www.graphpad.com/ | |
| Software, algorithm | Imaris | Bitplane | https://imaris.oxinst.com/ | |
| Software, algorithm | Adobe Illustrator | Adobe | https://www.adobe.com/ | |
| Software, algorithm | Loupe Browser 6.0 | 10× Genomics, Cell Ranger | https://www.10xgenomics.com | |
| Software, algorithm | Loupe VDJ Browser 4.0 | 10× Genomics, Cell Ranger | https://www.10xgenomics.com | |
| Software, algorithm | ImageJ2 | ImageJ | https://imagej.nih.gov/ij/ | |

## Mice

C57BL/6 (B6, WT) mice were purchased from The Jackson Laboratory. *Bcl6<sup>fl/fl</sup>* (*Hollister et al., 2013*) and *Foxp3-Yfp<sup>cre</sup>* mice (*Rubtsov et al., 2008*) were crossed in Alexander Dent's Lab (*Wu et al., 2016*). NZM2328 mice (*Jacob et al., 2003*) were a gift to Michelle Kahlenberg from Chaim Jacob (USC). All mice were bred and maintained under specific pathogen-free conditions. Relevant mice were inter-bred to obtain *Bcl6<sup>fl/fl</sup> Foxp3-Yfp<sup>cre</sup>* mice. All the animal experiments were conducted in compliance with the protocols reviewed and approved by the Institutional Animal Care and Use Committee of the University of Michigan.

## Antigen preparation
### DEL purification
Duck eggs were locally purchased and DEL was purified as previously described (*Allen et al., 2007*). Four-hundred ml of duck egg whites were blended with 2 l of buffer A (0.1 M ammonium acetate buffer, pH = 9), filtered through two layers of Kim wipes, and stirred overnight (O/N) at 4°C with 4.5 g CM Sephadex C-25 beads (GE Healthcare) preequilibrated for 3 hr with 100 ml of buffer A. The beads were then transferred into Buchner funnel covered with seven layers of kimwipes, washed with 1 l of buffer A and eluted with 200–300 ml of buffer B (0.4 M ammonium carbonate buffer, pH = 9.2). The eluent was lyophilized, resuspended in 15 ml buffer A, and dialyzed two times in buffer A at 4°C. Insoluble material was removed by centrifugation for 5 min at 1000 rcf, and the supernatant was concentrated using a Centriprep filter (Millipore, MWCO 10,000 kDa) to 3 ml for loading onto a 1×66 cm gel filtration column comprised of 4 g of degassed Sephadex G-50 beads (GE Healthcare) preequilibrated in buffer A O/N. The column was washed with 200 ml of degassed buffer A and the concentrated eluent was then separated on the column using degassed buffer A to collect ten 5 ml

fractions. Eluted fractions were analyzed by SDS-PAGE and stained with Coomassie brilliant blue. DEL-containing (band at ~14 kDa) fractions were combined. DEL concentration was determined by SDS-PAGE using a standard curve with Hen egg lysozyme (Sigma).

## Generation of SA-DEL, SA-NucPr, and SA-OVA

SA-DEL was generated as previously described (*Turner et al., 2017b*). Purified DEL was conjugated to biotin at a 1:2 molar ratio using Sulfo-NHS-LC-Biotin (Thermo Fisher Scientific) according to the manufacturer's directions and incubated for 3 hr on ice. After dialyzing three times in PBS DEL-biotin incubated with SA (Sigma) at a 10:1 molar ratio for 30 min on ice, followed by removal of unbound DEL-bio by passage through a 30 kDa molecular weight cut-off desalting column (Bio-Rad). Nucleosome, RNP/Sm, Jo-1, Scl-70, and Ro (SSA) (AROTEC DIAGNOSTICS) were conjugated to biotin at a 1:50 molar ratio using Sulfo-NHS-LC-Biotin and incubated for 3 hr on ice. After dialyzing three times in PBS, nucleosome-biotin, RNP/Sm-biotin, Jo-1-biotin, Scl-70-biotin, and Ro (SSA)-biotin were incubated with SA at a 1:1 molar ratio for 30 min on ice. Antigen conjugation was verified by SDS gel electrophoresis. After incubation, antigens were aliquoted and kept at –20°C. OVA-bio (Thermo Fisher, NC0887816) were incubated with SA at a 4:1 molar ratio for 30 min on ice. All antigens were aliquoted and kept at –20°C.

## Generation of αIgM-SA, αIgM-SA-DEL, αIgM-SA-NucPrs

To generate anti-IgM-NucPr, SA, anti-IgM-bio, nucleosome-bio, Jo-1-bio, Scl-70-bio, RO (SSA)-bio, and RNPsm-bio were incubated at a 1:2:0.5:0.5:0.5:0.5:0.5 molar ratio for 30 min on ice. To generate anti-IgM-DEL, SA, anti-IgM-bio, and DEL-bio were incubated at a 1:2:2 molar ratio for 30 min on ice. To generate anti-IgM-SA, SA and anti-IgM-bio were incubated at a 1:2 molar ratio for 30 min on ice.

## Immunization

In some experiments mice were subcutaneously (s.c.) immunized with 50 µg SA-DEL in Ribi, reimmunized with 10 µg SA, or with SA-OVA or SA-NucPrs (with 10 µg of SA in each conjugate prep) in Ribi at day 8 after immunization. In some experiments serum from mice immunized with 50 µg SA in Ribi at day 8 after immunization were collected and injected into unimmunized mice followed by immunization with 10 µg SA or SA-NucPrs (containing 10 µg SA). In some experiments naïve mice were immunized with 10 µg SA or SA-NucPrs (containing 10 µg SA) directly. Lymphoid cells from the draining lymph nodes and the spleens of immunized mice were analyzed at the indicated time points. Blood was collected into Microvette CB 300 tube via cardiac puncture when the mice are under deep anesthesia. About 0.5–1 ml blood was obtained from one mouse. Serum recovered after centrifugation at 10,000 × *g* 5 min 20°C.

## Flow cytometry analyses and FACS sorting

Single-cell suspensions from draining lymph nodes and spleens were prepared and filtered through a 70 µm nylon cell strainer (BD). Red blood cells were lysed. Cells were washed in FACS buffer (2% FBS, 1 mM EDTA, 0.1% NaN$_3$ in PBS) and followed by surface staining for the indicated markers for 20 min at 4°C. NP-specific B cells or SA-specific B cells were detected with BCR-specific binding with NP-PE (Biosearch Technologies) or SA-PE (BioLegend). For intracellular staining, the FoxP3 intracellular staining kit (Thermo Fisher Scientific) was used according to the manufacturer's instructions. Samples were then incubated with anti-FoxP3, CTLA4, granzyme B, or Ig (H+L). All samples were acquired on a BD FACSCanto flow cytometer. For cell sorting, enriched B cells and T cells were incubated with Abs in Sorting buffer (0.5% FBS and 2 mM EDTA in PBS) and were performed on a BD FACSAria III cell sorter. All data were analyzed with FlowJo (version 10.6.0) software.

## ELISA

SA-specific IgG were detected in serum from blood by ELISA. Nunc 96-well ELISA plates were coated with 50 µl of 2 µg/ml SA (Sigma) in borate saline buffer (100 mM boric acid, 0.9% NaCl, pH = 7.4) overnight at 4°C. Wells were blocked with 0.1% Bio-Rad Gelatin in PBS plus 0.05% Tween-20. Twofold diluted serum samples were loaded into the plate. ELISA plates were incubated for 1 hr at room temperature. Plates were washed with PBS containing 0.05% Tween-20. Bound Ab detected with 1.5 µg/ml IgG-HRP (Invitrogen). After washing, the color was developed with TMB (Thermo Fisher).

The chromogenic reaction was stopped with 2N sulfuric acid and the plates were read with a Synergy HT microplate reader (Bio-Tek Incorporated) at 405 and 630 nm. All plates contained serial dilutions of the serum that was used to generate the calibration curve for quantitative comparison of the samples.

Anti-RNP was detected in female 52-week-old C57Bl/6 and 32- to 52-week-old NZM2328 mice (harvested at time of nephritis or at 52 weeks) via ELISA (Alpha Diagnostics, San Antonio, TX, USA) according to the manufacturer's instructions.

## Immunofluorescence staining

Freshly isolated lymph nodes were fixed in 1% PFA for 1 hr at room temperature, washed with PBS three times, and stored at 4°C in 30% sucrose solution overnight. Fixed samples were then transferred to OCT (Tissue-Tek) and snap-frozen. Thirty μm thick sections were cut via cryostat (Leica). The sections were dried at room temperature for 3 hr. They were blocked using normal rabbit serum (Sigma-Aldrich) in PBS with 0.2% Triton X-100 and then stained with anti-mouse CD3 PE-CF594 (BD) and anti-mouse Bcl6 Alexa Fluor 488 (BD) overnight. After washing, slides were then mounted in fluoromount-G (Southern Biotech) and analyzed via confocal microscopy with Leica SP5 II (Leica Microsystems) using a 25× objective. Images were processed using Imaris and ImageJ. For quantitative analysis of GCs, Bcl6-positive areas in B cell follicles were identified, manually outlined in ImageJ, and their area was then calculated.

## 10× Genomics and TCR immunorepertoire analysis

B6 mice were s.c. immunized with SA-DEL in Ribi and at day 8 were s.c. reimmunized with SA, SA-NucPr. Follicular T cells (CD4$^+$PD1$^{hi}$CXCR5$^{hi}$) were sorted from mice at day 11 after immunization into PBS + 2% FBS. Single-cell suspensions were subjected to counting and viability checks on the LUNA Fx7 Automated Cell Counter (Logos Biosystems) and diluted to a concentration of 700–1000 cells/μl. Single-cell libraries were generated using the 10× Genomics Chromium Controller with Immune Profiling reagents following the manufacturer's protocol (10× Genomics). Final library quality was assessed using the LabChip GX (PerkinElmer). Libraries were subjected to paired-end sequencing according to the manufacturer's protocol (Illumina NovaSeq 6000). Bcl2fastq2 Conversion Software (Illumina) was used to generate de-multiplexed Fastq files and the CellRanger Pipeline (10× Genomics) was used to align reads and generate count matrices. Barcodes with mitochondrial reads >5% were removed. Preprocessing, clustering, and dimensionality reduction were performed using Loupe Browser and Loupe V(D)J browser (10× Genomics, Cell Ranger). Graph-based clustering of Tfr were visualized in 2D using UMAP algorithm.

For human scRNA-seq analysis, publicly available datasets were downloaded for lymph nodes cells from SARS-CoV2 mRNA-vaccinated patients (GSE195673) (*Kim et al., 2022*). Seurat v4 (*Hao et al., 2021*) was used for downstream analysis based on Seurat vignettes. Cells which had less than 200 or more than 8000 transcripts and mitochondrial genes representing greater than 15% of total transcripts were removed. All remaining cells were then clustered and projected into UMAP plots. The optimal number of PCs used for UMAP dimensionality reduction was determined using Jackstraw permutations which resulted in the first 15 PCAs being used. Identification of follicular T cells was accomplished by adding a module score in Seurat. Clusters in which the module score was significant enriched for were identified as being follicular. Follicular cells were then reclustered and differentially gene expression analysis between each cluster and all other cells was performed.

## Analysis of the human B cells and Tfr conjugate formation

Human blood was collected with informed consent from healthy donors in accordance with a University of Michigan IRB approved protocol (HUM0007150). Fresh blood was subjected to centrifugation over a Ficoll-Paque Plus (GE Healthcare) density gradient, washed twice in PBS and resuspended cells in PBS. PBMC were stained with fluorophore-conjugated Abs. After washing, CD3$^-$CD19$^+$CD27$^-$ B cells and CD4$^+$CD19$^-$CD45RA$^{lo}$CXCR5$^+$CD25$^{hi}$CD127$^{lo}$ Tfr were sorted and resuspended in RPMI 1640 medium (GIBCO) with 10% FBS (Hyclone), 1% HEPES (Hyclone), 100 U/ml penicillin and 100 μg/ml streptomycin (GIBCO). 5×10$^4$ B cells were first incubated with 5 μg/ml anti-IgM complexes and 1 μg/ml recombinant human CD40L (BioLegend) for 30 min and then 1×10$^4$ Tfr were added. After incubation at 37 °C and 5% CO$_2$ for 36 hr, the cells were stained and analyzed by FACS. The ratio of CD3$^+$CD19$^+$ cells and CD3$^+$ cells were calculated.

## Statistics

Statistical tests were performed as indicated using Prism 8 (GraphPad). No blinding or randomization was performed for animal experiments, and no animals or samples were excluded from analysis. All the statistical details of experiments and statistical analysis can be found in figure legends. Differences between groups not annotated by an asterisk did not reach statistical significance. Outliers were not excluded. Of note, outlier experimental data points in several figures did not affect statistical significance of the data.

## Acknowledgements

We thank Prof. Jason Cyster (UCSF) for useful discussions and comments on the manuscript. Single-cell processing and next-generation sequencing was carried out in the Advanced Genomics Core at the University of Michigan. Supported by the National Institute of Health R01 AI106806 and R21 AI142032 (IG), R01 AI132771 (ALD), R01-AR071384 (JMK), K24-AR076975 (JMK), P30-AR075043 (JMK, MPM).

## Additional information

### Competing interests

Joanne Michelle Kahlenberg: JMK has received Grant support from Q32 Bio, Celgene/BMS, Ventus Therapeutics, and Janssen. JMK has served on advisory boards for AstraZeneca, Eli Lilly, GlaxoSmith-Kline, Bristol Myers Squibb, Avion Pharmaceuticals, Provention Bio, Aurinia Pharmaceuticals, Ventus Therapeutics, Vera Therapeutics, and Boehringer Ingelheim. The other authors declare that no competing interests exist.

### Funding

| Funder | Grant reference number | Author |
| --- | --- | --- |
| National Institute of Allergy and Infectious Diseases | AI142032 | Irina L Grigorova |
| National Institute of Health | R21 AI142032 | Irina L Grigorova |
| National Institute of Health | R01 AI132771 | Alexander L Dent |
| National Institute of Health | R01-AR071384 | Joanne Michelle Kahlenberg |
| National Institute of Health | K24-AR076975 | Joanne Michelle Kahlenberg |
| National Institute of Health | P30-AR075043 | Joanne Michelle Kahlenberg Mitra P Maz |

The funders had no role in study design, data collection and interpretation, or the decision to submit the work for publication.

### Author contributions

Fang Ke, Zachary L Benet, Conceptualization, Investigation, Methodology, Writing – original draft; Mitra P Maz, Jianhua Liu, Investigation; Alexander L Dent, Joanne Michelle Kahlenberg, Methodology; Irina L Grigorova, Conceptualization, Supervision, Funding acquisition, Writing – original draft

### Author ORCIDs

Irina L Grigorova http://orcid.org/0000-0002-4963-7403

### Ethics

This study was performed in strict accordance with the recommendations in the Guide for the Care and Use of Laboratory Animals of the National Institutes of Health. All of the animals were handled according to approved institutional animal care and use committee (IACUC) protocols of the University

of Michigan. The protocol was approved by the Committee on the Ethics of Animal Experiments of the University of Minnesota (Protocol Number: PRO00009157).

### Decision letter and Author response
Decision letter https://doi.org/10.7554/eLife.83908.sa1
Author response https://doi.org/10.7554/eLife.83908.sa2

---

## Additional files

### Supplementary files
• Supplementary file 1. Table 1: Significant genes in the clusters 1–12 of murine follicular T cells. Single-cell 10× Genomics (Cell Ranger) analysis of gene expression in *Foxp3* expressing CD4$^+$CD8$^-$ B220$^-$CXCR5$^{hi}$PD1$^{hi}$ cells sorted from the draining inguinal lymph nodes (dLNs) of mice treated as described in *Figure 1A*. Follicular regulatory T cells (Tfr) were reclustered in Loupe software (Cell Ranger) based on gene expression into clusters 1–12 (see *Figure 2*). Statistical significance of mean gene expression in Tfr cluster compared to other Tfr was calculated based on negative binomial test (Cell Ranger).

• MDAR checklist

### Data availability
10x single cell data has been deposited in GEO under the accession code GSE216236.

The following dataset was generated:

| Author(s) | Year | Dataset title | Dataset URL | Database and Identifier |
|---|---|---|---|---|
| Ke F, Grigorova I | 2022 | Gene expression and VDJ TCR profile of follicular T cells from immunized with nuclear proteins and control mice | https://www.ncbi.nlm.nih.gov/geo/query/acc.cgi?acc=GSE216236 | NCBI Gene Expression Omnibus, GSE216236 |

The following previously published dataset was used:

| Author(s) | Year | Dataset title | Dataset URL | Database and Identifier |
|---|---|---|---|---|
| Kim W, Zhou JQ, Horvath SC, Schmitz AJ, Sturtz AJ, Lei T, Liu Z, Kalaidina E, Thapa M, Alsoussi WB, Haile A, Klebert MK, Suessen T, Parra-Rodriguez L, Mudd PA, Whelan SJ, Middleton WD, Teefey SA, Pusic I, O'Halloran JA, Presti RM, Turner JS, Ellebedy AH | 2022 | Germinal centre-driven maturation of B cell response to SARS-CoV-2 mRNA vaccination | https://www.ncbi.nlm.nih.gov/geo/query/acc.cgi?acc=GSE195673 | NCBI Gene Expression Omnibus, GSE195673 |

---

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
