## [Editor Report]

It is well known that Tfr cells have the capacity to preferentially suppress autoimmune antibody responses, but it is not known why such specificity is generated. This important study provides new information as to how self-reactive antibody responses are regulated and has significant implications for the fields of autoimmunity and vaccine design. The authors added careful controls which are convincing enough.

---

## [Decision Letter]

**Decision letter after peer review:**

Thank you for submitting your article "Nuclear proteins acquiring Germinal Center B cells are specifically suppressed by follicular regulatory T cells." for consideration by *eLife*. Your article has been reviewed by 2 peer reviewers, and the evaluation has been overseen by a Reviewing Editor and Betty Diamond as the Senior Editor. The following individual involved in review of your submission has agreed to reveal their identity: James Wing (Reviewer #1).

Essential revisions:

1) Experiments should be performed more stringently. For experiments of primary-boost immunizations, the different groups should be boosted with same amount (molar ratio) of SA. This requires the authors to determine the amounts of SA after different conjugations. Given SA-NucPr and SA-OVA were conjugated at different ratios (1:1 vs 4:1), the levels of conjugations must be properly evaluated. In addition, please include an additional group without SA boosting as a control.

2) In Figure S3A, SA-NucPr boosting immunization appeared to disrupt the GC architecture. Would the authors please provide whole tissue sections and more details of description and quantification? In Single cell analyses, could authors also analyze the changes in Tfh cells by SA-NucPr?

3) According to reviewer 1's comments, appropriate editorial changes should be done.

*Reviewer #1 (Recommendations for the authors):*

Some other issues also need clarification.

Figure 1A: Are Tfh percentages affected?

Supplementary figure 2C it would be helpful to display genes with links to previous Tfr literature such as Neuritin, IL1R2, IL1RN. Genes such as CD40LG, IL21, CXCL13 etc might also help resolution since they would be expected to be concentrated in Tfh. Confirmation of expression of the genes used in the modular score is also important modules are often contaminated by cells expressing only some genes that may resolve in later clustering.

Figure 2I may be better represented as % positive since the CD25 expression is bi-modal.

Figure 3H: Are the PB and GCPB gating strategies exclusive of one another? It seems possible that the PB will contain a large proportion of GCPB.

Figure 4: Did the authors examine Tfr in the NZM2328 mice? Given the focus of the paper it seems strange to omit this.

Figure 5: Please provide information as to the proportion and numbers of Tfh.

Figure 6A: The authors look at the SA-NucPr response in previously unimmunized mice at day 3. Would this not be too early to see a clear GC response? Can the authors more clearly explain the rationale behind this experiment?

Figure 7B: The gating definition of doublets and singlets should be shown. Are the authors including the CD3+CD19^+^ cells in their singlet population, what do they consider these to be?

A few typos need correction e.g. page 6 line 12 "TGFbetta1".

Unusual phrasing or grammatical errors are common e.g. page 3 line 27 "..presenting cognate to them self-antigens in the immunoregulation"

The title is worded strangely as it currently seems to imply that nuclear proteins acquire germinal center B-cells. Maybe "acquiring" could be replaced with "specific" or "recognizing".

Including the s in the Tfr abbreviation (Tfrs) is not needed and inconsistent with the wider literature.

*Reviewer #2 (Recommendations for the authors):*

Experiments should be performed more stringently. For experiments of primary-boost immunizations, the different groups should be boosted with same amount (molar ratio) of SA. This requires the authors to determine the amounts of SA after different conjugations. Given SA-NucPr and SA-OVA were conjugated at different ratios (1:1 vs 4:1), the levels of conjugations must be properly evaluated.

In addition, please include an additional group without SA boosting as a control.

In Figure S3A, SA-NucPr boosting immunization appeared to disrupt the GC architecture. Would the authors please provide whole tissue sections and more details of description and quantification?

In Single cell analyses, could authors also analyze the changes in Tfh cells by SA-NucPr?

---

## [Author Response]

Essential revisions:1) Experiments should be performed more stringently. For experiments of primary-boost immunizations, the different groups should be boosted with same amount (molar ratio) of SA. This requires the authors to determine the amounts of SA after different conjugations. Given SA-NucPr and SA-OVA were conjugated at different ratios (1:1 vs 4:1), the levels of conjugations must be properly evaluated. In addition, please include an additional group without SA boosting as a control.

We thank the reviewers for noticing this point! It is indeed very important that boosting immunizations contain an equivalent amount of SA. This is exactly how the experiments were planed and performed. The mistake was in the methods description and is now corrected.

We performed additional set of experiments to compare SA-DEL immunized mice boosted with SA or SA-NucPr to not boosted control mice (See new data in Figure 3- supplement 1C-E).

2) In Figure S3A, SA-NucPr boosting immunization appeared to disrupt the GC architecture. Would the authors please provide whole tissue sections and more details of description and quantification?

In the revised version of the manuscript we included immunofluorescent analysis of multiple tissue sections from multiple mice and quantified the size of GCs based on the outline of the Bcl6^+^ cells in the follicles within one cluster. Quantification of the GC area was performed in ImageJ and is consistent with decreased size of GC in mice boosted with SA-NucPrs compared to SA (see updated Figure 3- supplement 1A). We do not see immediate evidence for disruption of GC architecture, but more in-depth analysis will be performed in the future studies.

In Single cell analyses, could authors also analyze the changes in Tfh cells by SA-NucPr?

The analysis of the statistically significant changes in Tfh cells gene expression is now included in Figure 2- supplement 1G. The single cell data for two selected genes of interest are illustrated in Figure 2- supplement 1H.

3) According to reviewer 1's comments, appropriate editorial changes should be done.

We have also tried to address most of the additional points raised by the reviewers, as indicated below.

Reviewer #1 (Recommendations for the authors):Some other issues also need clarification.Figure 1A: Are Tfh percentages affected?

Please see added Figure 1E

Supplementary figure 2C it would be helpful to display genes with links to previous Tfr literature such as Neuritin, IL1R2, IL1RN. Genes such as CD40LG, IL21, CXCL13 etc might also help resolution since they would be expected to be concentrated in Tfh. Confirmation of expression of the genes used in the modular score is also important modules are often contaminated by cells expressing only some genes that may resolve in later clustering.

Thank you for raising this point!

We have now analyzed the expression of suggested genes in the follicular-like T cell clusters from human LNs and added *CD40LG, IL21, CXCL13* to Figure 2- supplement 2C. Unfortunately expression of *Neuritin, IL1R2, IL1RN* was not detectable in the analyzed dataset.

Additional discussion of the analysis is now included in the manuscript.

Figure 2I may be better represented as % positive since the CD25 expression is bi-modal.

Thank you! We reanalyzed the data and substituted for % of CD25 positive cells.

Figure 3H: Are the PB and GCPB gating strategies exclusive of one another? It seems possible that the PB will contain a large proportion of GCPB.

The PB and GCPB gates are not exclusive. PB gate does contain GCPB subset.

Figure 4: Did the authors examine Tfr in the NZM2328 mice? Given the focus of the paper it seems strange to omit this.

We have now added analysis of Tfr cells in the NZM2328 mice into the manuscript (Figure 4F).

We found increased variability of Tfr cells in different sets of NZM2328 mice. However, in 3 independent experiments SA-NucPr boosting led to increased numbers and frequency of Tfr cells as compared to SA boosting.

Figure 5: Please provide information as to the proportion and numbers of Tfh.

Added to Figure 5D, E.

Figure 6A: The authors look at the SA-NucPr response in previously unimmunized mice at day 3. Would this not be too early to see a clear GC response? Can the authors more clearly explain the rationale behind this experiment?

In the original experiments we observed an increase in Tfr cells at 3 days after the boost of SA-DEL immunized mice with SA-NucPrs. Therefore we have asked whether SANucPr immunization in the absence or presence of SA-specific Abs may be sufficient to promote Tfr response in 3 days. In this set of experiments analysis of GCs is secondary to the analysis of Tfr cells.

We agree with the reviewer that at 3 days post immunization foreign antigen-specific GC response would not be significant if any. However, small GCs are present in ILN in mice (under non-germ free conditions) even in the absence of immunization with foreign antigens. Therefore, if SA-NucPr promoted Tfr response, then some effects on the pre-existing GCs could have been expected.

Figure 7B: The gating definition of doublets and singlets should be shown. Are the authors including the CD3+CD19^+^ cells in their singlet population, what do they consider these to be?

We have included the gating strategy for singlets and doublets in Figure 7B.

There are very few CD3+CD19^+^ cells that fall into the singlets gate (possibly due to the smaller size of some cells in doublets or due to nonspecific staining of few cells).

A few typos need correction e.g. page 6 line 12 "TGFbetta1".Unusual phrasing or grammatical errors are common e.g. page 3 line 27 "..presenting cognate to them self-antigens in the immunoregulation"The title is worded strangely as it currently seems to imply that nuclear proteins acquire germinal center B-cells. Maybe "acquiring" could be replaced with "specific" or "recognizing".Including the s in the Tfr abbreviation (Tfrs) is not needed and inconsistent with the wider literature.

Corrected.

Reviewer #2 (Recommendations for the authors):Experiments should be performed more stringently. For experiments of primary-boost immunizations, the different groups should be boosted with same amount (molar ratio) of SA. This requires the authors to determine the amounts of SA after different conjugations. Given SA-NucPr and SA-OVA were conjugated at different ratios (1:1 vs 4:1), the levels of conjugations must be properly evaluated.

Thank you for this comment! When planning the experiments, we also wanted to ensure similar acquisition of SA by GC B cells. Because of that, all experiments have been performed using 10 ug of SA whether unbound or in conjugates with NucPrs or OVA. Unfortunately we have made a mistake in the methods description which is corrected in this revision.

In addition, please include an additional group without SA boosting as a control.

These experiments were repeated and are added to Figure 3- supplement 1C-E

In Figure S3A, SA-NucPr boosting immunization appeared to disrupt the GC architecture. Would the authors please provide whole tissue sections and more details of description and quantification?

Please see changes to the Figure 3- supplement 1A, B.

In Single cell analyses, could authors also analyze the changes in Tfh cells by SA-NucPr?

Please see Figure 2- supplement 1G, H.